# ViT-PSO-SVM: Cervical Cancer Predication Based on Integrating Vision Transformer with Particle Swarm Optimization and Support Vector Machine

**DOI:** 10.3390/bioengineering11070729

**Published:** 2024-07-18

**Authors:** Abdulaziz AlMohimeed, Mohamed Shehata, Nora El-Rashidy, Sherif Mostafa, Amira Samy Talaat, Hager Saleh

**Affiliations:** 1College of Computer and Information Sciences, Imam Mohammad Ibn Saud Islamic University (IMSIU), Riyadh 13318, Saudi Arabia; aialmohimeed@imamu.edu.sa; 2Bioengineering Department, Speed School of Engineering, University of Louisville, Louisville, KY 40292, USA; 3Machine Learning and Information Retrieval Department, Faculty of Artificial Intelligence, Kafrelsheiksh University, Kafrelsheiksh 13518, Egypt; noura.alrashidy@ai.kfs.edu.eg; 4Faculty of Computers and Artificial Intelligence, South Valley University, Hurghada 84511, Egypt; sherif.mostafa@fcih.svu.edu.eg; 5Computers and Systems Department, Electronics Research Institute, Cairo 12622, Egypt; amtalat@yahoo.com; 6Insight SFI Research Centre for Data Analytics, Galway University, H91 TK33 Galway, Ireland; 7Research Development, Atlantic Technological University, Letterkenny, H91 AH5K Donegal, Ireland

**Keywords:** cervical cancer, diagnostic model, ViT-PSO-SVM

## Abstract

Cervical cancer (CCa) is the fourth most prevalent and common cancer affecting women worldwide, with increasing incidence and mortality rates. Hence, early detection of CCa plays a crucial role in improving outcomes. Non-invasive imaging procedures with good diagnostic performance are desirable and have the potential to lessen the degree of intervention associated with the gold standard, biopsy. Recently, artificial intelligence-based diagnostic models such as Vision Transformers (ViT) have shown promising performance in image classification tasks, rivaling or surpassing traditional convolutional neural networks (CNNs). This paper studies the effect of applying a ViT to predict CCa using different image benchmark datasets. A newly developed approach (ViT-PSO-SVM) was presented for boosting the results of the ViT based on integrating the ViT with particle swarm optimization (PSO), and support vector machine (SVM). First, the proposed framework extracts features from the Vision Transformer. Then, PSO is used to reduce the complexity of extracted features and optimize feature representation. Finally, a softmax classification layer is replaced with an SVM classification model to precisely predict CCa. The models are evaluated using two benchmark cervical cell image datasets, namely SipakMed and Herlev, with different classification scenarios: two, three, and five classes. The proposed approach achieved 99.112% accuracy and 99.113% F1-score for SipakMed with two classes and achieved 97.778% accuracy and 97.805% F1-score for Herlev with two classes outperforming other Vision Transformers, CNN models, and pre-trained models. Finally, GradCAM is used as an explainable artificial intelligence (XAI) tool to visualize and understand the regions of a given image that are important for a model’s prediction. The obtained experimental results demonstrate the feasibility and efficacy of the developed ViT-PSO-SVM approach and hold the promise of providing a robust, reliable, accurate, and non-invasive diagnostic tool that will lead to improved healthcare outcomes worldwide.

## 1. Introduction

Cervical cancer (CCa) poses significant challenges for women, with increasing incidence and mortality rates in recent years. It is currently the fourth-most prevalent cancer in women worldwide with over 660,000 newly diagnosed cases and roughly 350,000 deaths in 2022 mainly impacting those in developing and low-income nations owing to the shortage of early detection methods [1]. Various techniques, including visual inspection, Papanicolaou (Pap) tests, histopathology tests, and human papillomavirus (HPV) testing, are employed for CCa detection [2]. However, the availability of these tests may be limited in certain countries. For instance, although the HPV test is susceptible, it can yield false negatives, potentially overlooking early stage HPV infections. Furthermore, HPV infections are more common in younger women but are more likely to persist in older women. Therefore, it is crucial to consider screening methods that account for social, biological, and ethical factors, as early detection of CCa improves prognosis. Hence, the development of an accurate system capable of early symptom detection is vital [3,4,5].

Medical imaging tools involving ultrasound, computed tomography (CT) scans, and magnetic resonance imaging (MRI) reveal extensive information about infected tissues and tumor features like dimensions, position, spread, and growth rate [6,7,8]. However, the full benefit of these images is often not realized due to a shortage of professionals, which leads to diagnostic subjectivity. Therefore, there is a critical need to integrate advanced technology, such as artificial intelligence (AI), with medical images to develop robust and generalized models that can provide accurate objective diagnoses. Both of Deep learning (DL) and machine learning (ML) were substantially enhanced several sectors, including healthcare, finance, and even healthcare [9].

When it comes to CCa, computer vision techniques, such as convolutional neural networks (CNNs), can learn abstract features from images, which can then be utilized to build medical Computer-Aided Diagnosis (CAD) systems [10]. However, traditional CNN models have limitations that impact their performance, especially when dealing with complex data [3]. For instance, CNNs tend to focus on local patterns and rely solely on weight sharing within the same layer, which restricts their ability to understand global contexts and capture spatial dependencies [7]. This returns to several reasons including the following: (1) ViT can capture the global context of data in contrast to traditional CNN, which can process the local context of data; (2) ViT also can employ self-attention, which allows it to capture and understand the relations and the long-term dependencies; (3) ViT is a highly scalable model which makes it suitable for high resolution without the need for resizing or cropping; and (4) ViT provides strong transfer learning, which allows it to fine-tune and converge faster. Transfer learning provides a promising approach by leveraging pre-trained deep learning models, particularly CNNs. This approach allows for the transfer of previously learned representations to new tasks, thereby overcoming the limitations associated with traditional CNN models [11,12,13]. Revolutionary approaches, such as Swin [14] and ViT [15] Transformers, have been developed. ViT Transformers, in particular, utilize self-attention mechanisms to extract global dependencies. By leveraging self-attention mechanisms, there is no longer a need for manual feature engineering. This advancement provides promising techniques for image processing and classification, contributing to the development of more accurate and efficient diagnostic models [16,17].

A population-based metaheuristic model named particle swarm optimization (PSO) emerged upon observing the behavior of swarms. The way this algorithm works is by repeatedly looking for an optimal output. In this work, PSO is used to determine the best feature subset. This phase not only helps to make the proper decision based on the optimal feature subset, but it also reduces model complexity because it is dependent on the most impacted feature subset.

This paper’s primary goal is to present an expansive, reliable, and accurate framework for CCa classification. In addition to offering promising results in terms of classification accuracy, this model can identify the features who have the greatest impact based on the ViT Transformer and PSO metaheuristic optimization. The following points convey an outline of this paper’s primary contributions:Utilization of the ViT Transformer to capture both local and global contextual information from medical images, thereby improving the accuracy of image classification.Implementation of the PSO model for selecting an optimal feature subset and reducing feature complexity.Enhancement of the model’s performance by replacing the last layer with an SVM machine classifier, leveraging the strengths of both DL and SVM techniques.Conducting thorough comparison of the proposed model with other models, such as the ViT Transformer and pre-trained CNN models, to assess and verify its superiority in terms of functionality and efficacy.Creation of visual explanations for the proposed model predictions by superimposing the GradCAM heatmaps across the original cell images as an explainable AI tool.

Remaining parts of this research is structured into four sections: Section 2 covers the literature review, whereas Section 3 details the deployed methodology and dataset. Section 4 assesses the proposed technique, model results, and evaluation measures before concluding in Section 5.

## 2. Literature Reviews

Precancerous mutations usually give rise to cervical cancer over a period of 10 to 20 years. A cervical screening test is the only effective way to tell whether the cervix possesses abnormal cells which could lead to cervical cancer. Cervical cancer is often screened for through a Pap test. Because the Pap test can detect abnormal or precancerous alterations in the cervix’s cells, it is crucial for the early detection of cervical cancer. For this reason, Kyi et al. [18] employed a computer-assisted screening system of Pap smear pictures to construct a computer-assisted cervical cancer screening system. Researchers relied on a shape-based method of iteration to define nuclei via cell segmentation, combined with a marker-control watershed method for distinguishing between overlapping cytoplasm. Then, intriguing aspects of the fragmented nuclei and cytoplasm have been identified using feature extraction algorithms. Using the SipakMed and Herlev datasets, five classifiers’ results were combined to form the bagging ensemble classifier, which achieved 98.27% accuracy in two-class classification and 94.09% in five-class classification.

Rather than utilizing a microscope to identify lesion images, Wong et al. [19] searched for a method of intelligently analyzing samples. The authors set out to develop an AI image identification system that could utilize liquid-based Pap tests to recognize the extent of CCa lesions in accordance with the Bethesda classification of cancer. ResNet50V2 with ResNet101V2 represent two models developed by integrating DL with transfer learning methods. The assessment findings indicated that the ResNet50V2 model performed more effectively, for certain image categories classified with 98% precision and 97% accuracy.

Pallavi et al. [20] proposed a method utilizing adaptive fuzzy k means clustering to extract the ROI from cells with cytoplasm and nucleus segments from aberrant Pap smear images, making it useful for early CCa detection.

In order to save time and reduce the possibility of error, Wanli et al. [21] developed a DL-based framework for replacing the manual screening of cytopathology pictures for cervical cell classification tasks. In order to figure out the final classification based on CNN and Visual Transformer modules, the Multilayer Perceptron module is built to fuse the local and global data. The accuracy of the suggested framework, against combined CRIC and SipakMed datasets reached 91.72%. For the same reasons and via the same approach, two automatic CAD methods have been suggested by Maurya et al. [22]. Regarding the objective of classifying cervical cell Pap smear images, the first one employed an ensemble of CNNs and ViT networks, while the other used transfer learning with an LSTM and CNN framework. On the “SipakMed” Pap Smear dataset, the proposed algorithms ViT-CNN and CNN-LSTM achieved 95.80% and 97.65% accuracy, respectively. However, the CNN-LSTM technique exceeded ViT-CNN with regard to of computing resource efficiency.

Gurram et al. [23] suggest utilizing Pap smear images for identifying cervical cancers. For better feature extraction and classification, the proposed approach leveraged a CNN based on the ResNet50 architecture. Their technique obtained 97.5% accuracy on the SipakMed pap-smear image dataset, while the VGG 11 architecture achieved 92.2%.

Transformers impose minimum architectural assumptions regarding the size of the data being received. Because of this property, Bhaswati et al. [24] provided a cross-attention-based transformer technique which can deal with extremely large-scale inputs for the accurate classification of CCa in Pap smear images.

Jarbas et al. [25] suggested a discriminatory texture analysis method for Pap smear cell images classification based on the Papanicolaou tests. This method achieved 87.57% accuracy with an AUC around 0.8983 applying LDA and SVM, respectively. In [4], the authors applied different pre-trained models, namely InceptionResNetV2, VGG19, DenseNet201, and Xception, for classifying cervical images within the SipakMed dataset.

Yaman et al. [26] provided an exemplar pyramid deep feature extraction method for detecting CCa using cervical cells at Pap smear images. The proposed method used DarkNet19 and DarkNet53 transfer learning-based feature extraction based on Neighborhood Component Analysis (NCA) and SVM on SipakMed and Mendeley Liquid-Based Cytology (LBC) datasets, achieving 98.26% for accuracy.

Despite the promising diagnostic performance achieved by the aforementioned studies in the early detection of CCa, none of them investigated the integration of the ViT Transformer that captures both local and global contextual information from medical images with a PSO model that has the ability to select an optimal feature subset to reduce feature complexity and with an optimized SVM classification model thereby improving the diagnostic accuracy of image classification. As far as we are aware, the suggested method is the first of its type to combine ViT with PSO and SVM in order to integrate ViT with PSO and SVM to seek early and precise classification of CCa using two well-known datasets, namely SipakMed and Herlev.

## 3. Materials and Method

The primary steps involved in classifying cervical cells are depicted in Figure 1. The main objective of this study is to propose a novel approach (ViT-PSO-SVM) to enhance the results obtained by ViT based on integrating PSO, and SVM. First, the proposed framework extracts features from the ViT. Then, PSO is used to reduce the complexity of extracted features and optimize feature representation. Finally, a softmax classification layer is replaced with an SVM classification model.

### 3.1. Database Description

We performed our experiments using two cervical cell image datasets:SipakMed consists of 4049 images of cervical cells provided by Plissiti et al. [27], which is used to evaluate the proposed model. It is a balanced dataset and it includes five classes of Pap smear images which are superficial–intermediate, parabasal, koilocytes, dyskeratotic, and metaplastic. A sample image for each class is shown in Figure 2.A total of 917 images of cervical cells located by Herlev [28]. The images have been classified into regular and abnormal categories. Figure 3 displays several sample images from each class.

### 3.2. Image Preprocessing/Augmentation

Image augmentation involves making changes to an image in terms of color and position. Positional manipulation is achieved by altering the position of pixels, while color manipulation involves changing the pixel values. It includes techniques such as flipping, cropping, resizing, and noise injection [29]. These techniques contribute to improving the overall generalization performance of the model by exposing it to a wide variety of images during the training process.

Flipping comprises a horizontal flip that reverses the image’s left–right orientation. This can help the model develop the ability to recognize items independent of their left–right orientation, and vertical flip which reverses the image’s top-bottom orientation. This can help the model learn to recognize items regardless of their orientation up or down [30].Resize: Standardizes the size of images in the dataset, making them consistent for the model input. This can help in the efficient batching and processing of images.Randomly cropping the image can assist the model in learning to recognize objects independent of where they appear in the image [31]. This can be valuable for tasks where the item of interest is not perfectly aligned in the image.Normalization helps in speeding up the convergence of the training process by standardizing the input data.

### 3.3. State-of-the-Art Approaches

DenseNet is built upon the concept of Dense blocks, where each layer within a block is connected to all other layers within the same block. These connections are established through concatenation. Input for the current layer is a stack of the feature maps from the earlier layers, allowing for effective information flow. To downsample the feature maps, DenseNet incorporates transition layers that consist of a 1 × 1 normalization layer and an average pooling layer. These transition layers also help compress the information and increase the efficiency of the network. As a result, DenseNet demonstrates its capability to perform various computer vision tasks [32,33].The residual connections in ResNet are built on the concept of skip connections, which allow for direct connections to earlier layers, bypassing certain layers and creating shortcuts in the gradient flow. This approach highlights the distinction between residual mapping and identity mapping [34]. By learning the residuals, the network can focus on refining the mapping rather than relying solely on a complete transformation from scratch. The ResNet architecture consists of a series of residual blocks, each containing multiple convolutional layers with varying structures [35,36]. These differences in structure reduce the computational complexity of the models and restore the feature dimension. This enables the network to learn both shallow and deep features effectively.AlexNet leverages convolutional layers to derive spatial characteristics from input images. The convolution operation entails rolling a set of learnable filters across the input image or feature map and conducting element-wise multiplications followed by summations [37]. The output feature map is created by stacking these activations across spatial dimensions. AlexNet captures several degrees of image information through convolutional layers with varying filter sizes and channel counts. AlexNet employs the rectified linear unit (ReLU) activation function after each convolutional and fully connected layer to cope with the vanishing gradient problem [37]. AlexNet uses max-pooling layers to minimize the spatial dimensions of feature maps, lowering computational complexity while providing some translation invariance. AlexNet uses fully connected layers at the network’s end to carry out categorization based on previously learned features.VGG-16 architecture is a deep CNN built for image classification problems. It stacks a succession of small 3 × 3 convolutional layers having a fixed receptive field, before adding max-pooling layers that minimize spatial dimensions. The VGG-16 design typically has 16 layers, including 13 convolutional layers and three fully connected layers [38].

### 3.4. ViT Transformer

The Vision Transformer (ViT) is a Transformer-based architecture specifically designed for image classification tasks. The self-attention mechanism employed in ViT enables tokens to compare and capture dependencies and relationships among different regions within an image. The feed-forward network processes these tokens to extract high-level features [39].

In the ViT architecture (shown in Figure 4), the Transformer encoder takes in the token representations as input and produces encoded outputs. These outputs are then fed to the subsequent feed-forward network to further process the token representations into higher-level features. Finally, the Transformer encodes the generated tokens’ outputs, which are then passed through the classification head to predict the image label [15].

One major advantage of ViT is its ability to capture image data without prior knowledge about spatial hierarchies or handcrafted features. However, ViT faces limitations when dealing with large-sized images [40]. To address this challenge, techniques such as patch overlapping and hybrid architectures that combine CNNs and Transformers have been proposed to enhance ViT’s performance on larger images [41].

The Vision Transformer (ViT) architecture replaces the convolutional layers typically with a self-attention system. The input image is divided into a grid of non-overlapping patches, each of which is linearly projected into a lower-dimensional representation known as a token [42].

If the image dimensions are H × W and the patch size is P × P, the number of tokens is N=(H/P)×(W/P). The augmented tokens are then linearly projected into a higher-dimensional space, enabling them to preserve convoluted visual features. To capture global connections and dependencies between tokens, self-attention is paired with a feed-forward neural network technique applied to token embeddings. Given the input token embeddings X, the self-attention (SA) approach generates attention weights and employs them to generate context-aware representations [43]. It is composed of three linear transformations: query (Q), key (K), and value (V).

$\left[q,k,v\right]={\mathrm{xU}}_{\mathrm{qkx}}$ That $U_{\mathrm{akx}\;}\in R^{D\times 3D_{h}}$ and $x\in R^{N\times D}$ denote the projected token embedding where D is the dimensional of the projected space and weighted sum computed for all values vs. in the sequence. The attention weights $A_{ij}$ depend on the pairwise resemblance of the two components of the sequence, and their respective query $q_{i}$ and key $k_{j}$ representations are computed as a softmax of the scaled dot-product between queries and keys.
(1)$$A=\mathrm{softmax}\left({\mathrm{qk}}^{T}/\sqrt{D_{h}}\right)\;\mathrm{that}\;A\in R^{N\times N}$$
and this process is carried out individually by multiple attention heads, with the results concatenated and linearly projected to provide the final attention output [43].
(2)SA(z)=Av

Multihead self-attention (MSA) is an extension of SA in which k self-attention procedures known as “heads” run concurrently and project their concatenated results.
(3)$$\mathrm{MSA}(z)=\left[{\mathrm{SA}}_{1}\left(z\right);{\mathrm{SA}}_{2}\left(z\right);\cdots ;{\mathrm{Sa}}_{k}\left(z\right)\right]U_{\mathrm{msa}}$$

The final Transformer layer’s output is fed into a classification head, which consists of a softmax activation followed by a linear transformation. The final class probabilities for the image are accordingly generated [44].

### 3.5. Practical Swarm Optimization (PSO)

Swarm optimization is a meta-heuristic technique inspired by the collective behavior observed in colonies of organisms such as bees and ants [45]. It leverages the collaboration and information exchange within these colonies to explore solution spaces and find optimal solutions [46]. The algorithm consists of a population of individuals referred to as agents, which iteratively explore the problem space and update their positions based on their experiences. Swarm intelligence is founded on the concept that intelligent behavior at the group level emerges from the collective behavior of agents at the individual level. Each agent maintains its velocity and position, optimizing them based on the best solution found so far. This is achieved through iterative position updates aimed at searching the solution space for the optimal solution. Swarm intelligence offers several advantages for handling complex optimization problems. Its inherent nature allows for parallel exploration of the solution space, facilitating escape from local optima and enabling the discovery of global solution spaces. Furthermore, swarm optimization exhibits robustness, as it is capable of handling noise in uncertain environments [47].

Our paper utilizes PSO [48] for optimizing feature extraction from ViT. In PSO, each particle (agent) represents a point in the feature space. Let’s assume there are n parameters that require optimization, thus the position of a particle can be represented as:(4)$$x_{i}=\left[x_{i_{1}},x_{i_{2}},\ldots \ldots ,x_{i_{n}}\right]$$

The population size is denoted as m, which represents the number of candidate solutions:(5)$$x_{i}=\left[X_{1},X_{2,\ldots \ldots \ldots \ldots }\ldots ,X_{m}\right]$$

During the exploration of the search space to reach the optimal solution, each particle defines its trajectory. It iteratively updates its solution based on the following equation
(6)$$X_{i}\left(t+1\right)=V_{i}\left(t\right)+V_{i}\left(t+1\right)$$

Here, t represents the iteration number of the algorithm, and V_i is the velocity component that aggregates along the M dimensions. This vector controls the movement of the particle throughout the search space. The movement is determined by three aspects:Inertia: It prevents the particle from changing its direction drastically by preserving the previous direction.Cognitive component: It influences the particle to return to its previous solution.Social component: It determines the propensity of each particle towards the best solution. Considering the above parameters, the velocity of a particle can be defined as:(7)$$V_{i}\left(t+1\right)=V_{i}\left(t\right)+C_{1}\left(P_{i}+X_{i}\left(t\right)\right)R_{1}+C_{2}\left(g-X_{\left(t\right)}\right)R_{2}$$

Here, P_i represents the best solution obtained so far (local best), and g refers to the overall best solution obtained (global best). The acceleration coefficients, C_1 and C_2, are real values ranging from 0 to 4. R_1 and R_2 are two diagonal matrices randomly generated from numbers between 0 and 1. Consequently, the trajectory followed by the particles can be visualized as

Initialization stepfor each particle N, initialize the practice position $X_{i}\left(0\right)\forall \in 1:N$initialize the best solution according to the initial position $P_{i}\left(0\right)=X_{i}\left(0\right)$calculate fitness function for each agent (practical) $f\left(X_{i}\left(0\right)\right)$if $f\left(X_{j}\left(0\right)\right)\geq f\left(X_{i}\left(0\right)\right)\forall i\neq j\forall i\neq j$ intimate the global best as $g=X_{i}\left(0\right)$repeat this step until reaching the stopping criteria (Update step)updates the agent velocity according to the following equation (Equation (4))
(8)$$V_{i}\left(t+1\right)=V_{i}\left(t\right)+C_{i}\left(P_{i}+X_{i}\left(t\right)\right)R_{1}+C_{2}\left(g-X_{\left(t\right)}\right)R_{2}$$Update the garnet position according to the following equation (Equation (3))
(9)$$X_{i}\left(t+1\right)=X_{i}\left(t\right)+V_{i}\left(t+1\right)$$Evaluate the fitness function for each practice $f\left(X_{i}\left(t+1\right)\right)$if $f\left(X_{i}\left(t+1\right)\right)\geq f\left(P_{i}\right)$ make update, make the best equal $P_{i}\left(X_{i}\left(t+1\right)\right)$if $f\left(X_{i}\left(t+1\right)\right)\geq f\left(g\right)$ update the global best $g=\left(X_{i}\left(t+1\right)\right.$At the end of an iterative process, the optimized salutation is g

### 3.6. Proposed Model

The proposed model (ViT-PSO-SVM) for image classification integrates the use of ViT Transformer, swarm optimization, and SVM to enhance feature extraction and classification as shown in Figure 5. This is accomplished through the following steps as shown in Algorithm 1.
First, the ViT Transformer serves as the backbone of the model and is responsible for extracting local and global features from the images. With its powerful ability to capture contextual information and semantics, ViT divides the images into small patches and utilizes self-attention mechanisms to effectively learn meaningful features and understand local and global dependencies.Second, to further enhance the feature selection process, PSO is utilized as an optimization technique. PSO collaboratively explores the search space, allowing the model to search for optimal features. PSO is capable of finding the most informative and discriminative features, thereby enhancing the overall classification performance.Third, an SVM model replaced the role of the softmax activation function within the last layer. SVM was chosen due to its efficacy in multi-class classification problems and flexibility to handle high-dimensional data. By utilizing SVM, the model achieves robust and accurate classification.

**Algorithm 1:** Proposed work  
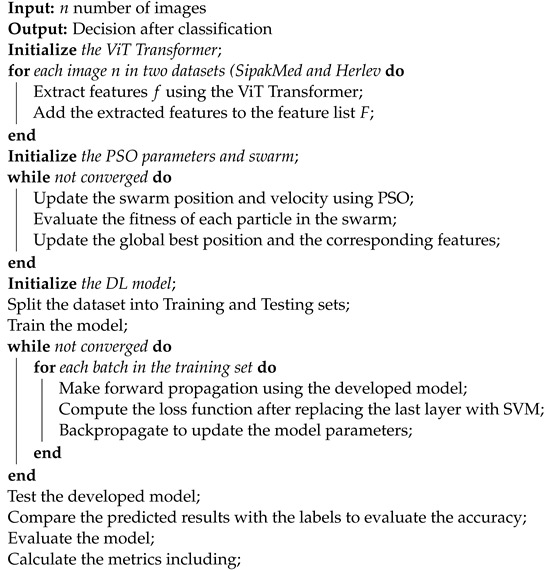


### 3.7. Explainable Artificial Intelligence (XAI)

Gradient-weighted Class Activation Mapping, or GradCAM, and XAI are both key concepts in the field of DL. XAI refers to the broader goal of boosting the transparency and understandability of AI systems for users. This includes methods that attempt to clarify how AI models make decisions, such as how they determine which predictions or classifications to make [49]. XAI aims to increase user comprehension and validation of the model’s behavior, improve the interpretability of AI systems, and promote a sense of confidence in systems. GradCAM is a particular XAI method that assists in the explanation of the model’s decision-making [50]. It achieves this by highlighting the regions of an input image that were responsible for the most influence on the prediction generated by the model. GradCAM does this by estimating the target class score’s gradients concerning the feature mappings [50]. The key areas of the input image are then highlighted in a heatmap made using these gradients. GradCAM facilitates human comprehension and interpretation of the model’s behavior by providing an animated illustration of the decision-making process. GradCAM indicates the input regions that the model depends on, which might assist in identifying possible shortcomings or biases in the model [50].

### 3.8. Model Evaluation

Where the correctly categorized positive class is termed as true positive (TP), the correctly classified negative class named true negative (TN), the wrongly classified positive class is labeled false positive (FP), and the incorrectly classified negative class is called false negative (FN).
(10)$$Accuracy=\frac{TP+TN}{TP+FP+TN+FN}$$
(11)$$Recall=\frac{TP}{TP+FN}$$
(12)$$Precision=\frac{TP}{TP+FP}$$
(13)$$F1-score=2\cdot \frac{precision\cdot recall}{precision+recall}$$

## 4. Experiments Results

This section presents the results of comparing the proposed model (Swin-GA-RF), the Swin Transformer, CNN, and pre-trained CNN models for classifying cervical cells. Several experiments and comparisons are conducted to validate the proposed model’s ability to enhance accuracy and generalization in classification. The model is trained on two datasets: SipakMed and Herlev, and performance is evaluated using metrics.

### 4.1. Experimental Setup

The Monai library, PyTorch, and Python were all utilized to form the models. Dataset split into 70%, 25% and 5% for training, testing and validation respectively. Table 1 depicts the number of images for SipakMed in each class while Table 2 displays the amount of images for Herlev in each class. A grid search was implemented to optimize the SVM hyper-parameters, using C = 0.1, gamma = 0.01 and kernel = poly.

#### 4.1.1. Setting of PSO

Table 3 provides an overview of the PSO characteristics used for selecting the optimal features. This paper’s primary objectives are to boost the outcomes and simplify the feature matrix. The size of the features in the last ViT layer before the output layer is 800 as shown in Table 4. Performance is increased and feature size is decreased whenever PSO is applied.

#### 4.1.2. Setting of Image Preprocessing/Augmentation

Image augmentation is beneficial in developing deep learning models, where large, diverse datasets are required to gain acceptable high accuracy. Image augmentation techniques enhance the diversity and variability of the training data. Table 5 shows the values of each transformation function that apply to each and Figure 6 shows the effect of each transformation function on the image.

### 4.2. The Results of SipakMed Images Dataset

This section presents the results of pre-trained CNN models, ViT Transformers, and the proposed models including their usage in different scenarios: binary classes, three classes, and five classes for the SipakMed dataset. The results showed that ViT-PSO-SVM demonstrated efficient and highest performance in extracting both local and global features using attention. POS helps reduce the complexity of features extracted from ViT, and an SVM is used instead of Softmax to make the final class prediction.

#### 4.2.1. The Results of Two Classes

Table 6 shows the results of the two classes with different models, including the pre-trained CNN models, ViT Transformer and the proposed models. From Table 6, we could make the following observations: (1) Among the models evaluated, DenseNet121 achieved an accuracy of 94.787. It demonstrated a precision of 94.923, a recall of 94.787, and an F1-score of 94.736. This model shows consistent performance across the precision, recall, and F1-score metrics. (2) VGG16, on the other hand, attained an accuracy of 91.200. With a precision of 91.200, recall of 91.200, and F1-score of 91.200, VGG16 exhibits slightly lower performance compared to other models. (3) ResNet18 showcases improved results, achieving an accuracy of 96.915. It demonstrates a precision of 96.916, a recall of 94.592, and an F1-score of 94.586. These metrics indicate notable performance in accurately classifying cervical cancer. (4) AlexNet achieved an accuracy of 95.957. With a precision of 95.957, recall of 95.957, and F1-score of 95.938, it demonstrates competitive performance among the evaluated models. This is due to the overlapping of the pooling layers which allows AlexNet to capture the important patterns. (5) The ViT Transformer model is considered a breakthrough by applying Transformer architecture. This enhancement achieved an accuracy of 98.126. It exhibits a precision of 98.127, a recall of 98.126, and an F1-score of 95.456. These results indicate consistent and reliable performance in cervical cancer classification. (6) Among all the models, the proposed models record the highest performance and the highest of the all is ViT-PSO-SVM at an accuracy of 99.112, a precision of 99.119, a recall of 99.112, and an F1-score of 99.113. This model exhibits excellent performance, surpassing the other evaluated models in accuracy and other performance metrics. These improvements can be attributed to the combination of multiple techniques: first, the ViT for extracting local and global features, then the utilization of the PSO to reduce feature complexity by selecting the most impacted features, and then SVM for giving the last decision in the developed model.

Based on time, AlexNet and ResNet18 are the quickest to process compared to the ViT Transformer and the proposed model, and the difference in time between them is insignificant. The main goal of this paper is to enhance the performance of models so that the ViT-PSO-SVM is recorded as having the highest performance.

#### 4.2.2. The Results of Three Classes Using SipakMed

Table 7 shows the results of the three classes with different models, including the pre-trained CNN models, ViT Transformer and the proposed models. From Table 7, we could make the following observations: (1) Among the models evaluated, DenseNet121 achieved an accuracy of 95.112. It demonstrated a precision of 95.115, a recall of 95.112, and an F1-score of 95.113. This model shows consistent performance across the precision, recall, and F1-score metrics. (2) VGG16, on the other hand, attained an accuracy of 93.199. With a precision of 93.187, recall of 93.199, and F1-score of 93.157, VGG16 exhibits slightly lower performance compared to other models. (3) ResNet18 showcases improved results, achieving an accuracy of 97.929. It demonstrates a precision of 97.934, a recall of 97.929, and an F1-score of 97.926. These metrics indicate notable performance in accurately classifying cervical cancer. (4) AlexNet achieved an accuracy of 94.688. With a precision of 94.896, recall of 94.688, and F1-score of 94.744, it demonstrates competitive performance among the evaluated models. (5) The ViT Transformer model is considered a breakthrough in applying Transformer architecture. This enhancement achieved an accuracy of 98.126. It exhibits a precision of 98.133, a recall of 98.126, and an F1-score of 98.126. These results indicate consistent and reliable performance in cervical cancer classification. (6) Among all the models, the proposed models record the highest performance and the highest of all is ViT-PSO-SVM with an accuracy of 99.211. It demonstrates a precision of 99.211, a recall of 99.211, and an F1-score of 99.211. This model exhibits excellent performance, surpassing the other evaluated models in accuracy and other performance metrics.

#### 4.2.3. The Results of Five Classes Using SipakMed

Table 8 shows the results of the five classes with different models, including the pre-trained CNN models, ViT Transformer and the proposed models. From Table 8 we could make the following observations: (1) Among the models evaluated, DenseNet121 achieved an accuracy of 91.362. It demonstrated a precision of 91.368, a recall of 91.362, and an F1-score of 91.357. This model shows consistent performance across the precision, recall, and F1-score metrics. (2) VGG16, on the other hand, attained an accuracy of 90.962. With a precision of 90.514, recall of 90.962, and F1-score of 90.101, VGG16 exhibits slightly lower performance compared to DenseNet121. This could be attributed to the architectural disparities of VGG 16, which affect the flow of information and the generalization ability of the model. (3) ResNet18 showcases improved results, achieving an accuracy of 94.592. It demonstrates a precision of 94.733, a recall of 94.592, and an F1-score of 94.586. These metrics indicate notable performance in accurately classifying cervical cancer. (4) AlexNet achieved an accuracy of 93.707. With a precision of 93.786, recall of 93.707, and F1-score of 93.693, it demonstrates competitive performance among the evaluated models. This is due to the overlapping of the pooling layers which allows AlexNet to capture the important patterns. (5) The ViT Transformer model is considered a breakthrough in CV by applying Transformer architecture. This enhancement achieved an accuracy of 95.477. It exhibits a precision of 95.482, a recall of 95.477, and an F1-score of 95.456. These results indicate consistent and reliable performance in cervical cancer classification. (6) Among all the models, the proposed models record the highest performance and the highest of all is ViT-PSO-SVM at an accuracy of 97.247. It demonstrates a precision of 97.253, a recall of 97.247, and an F1-score of 97.239. This model exhibits excellent performance, surpassing the other evaluated models in accuracy and other performance metrics. These improvements can be attributed to the combination of multiple techniques.

### 4.3. The Results of Herlev Images Dataset

Table 9 shows the results of the two classes with different models, including the pre-trained CNN models, ViT Transformer and the proposed models. From Table 9, we could make the following observations: (1) Among the models evaluated, DenseNet121 achieved an accuracy of 90.355. It demonstrated a precision of 90.967, a recall of 90.355, and an F1-score of 90.625. This model shows consistent performance across the precision, recall, and F1-score metrics. (2) VGG16, on the other hand, attained an accuracy of 89.285. With a precision of 89.34, recall of 89.285, and F1-score of 89.75, VGG16 exhibits slightly lower performance compared to other models. (3) ResNet18 showcases improved results, achieving an accuracy of 93.333. It demonstrates a precision of 93.889, a recall of 93.333, and an F1-score of 93.002. These metrics indicate notable performance in accurately classifying cervical cancer. (4) AlexNet achieved an accuracy of 91.251. With a precision of 91.769, recall of 91.251, and F1-score of 91.468, it demonstrates competitive performance among the evaluated models. This is due to the overlapping of the pooling layers which allows AlexNet to capture the important patterns. (5) The ViT Transformer model is considered a breakthrough by applying Transformer architecture. This enhancement achieved an accuracy of 95.238. It exhibits a precision of 95.28, a recall of 95.238, and an F1-score of 95.141. These results indicate consistent and reliable performance in cervical cancer classification. (6) Among all the models, the proposed models record the highest performance and the highest of all is ViT-PSO-SVM at an accuracy of 97.778, a precision of 97.949, a recall of 97.778, and an F1-score of 97.805. This model exhibits excellent performance, surpassing the other evaluated models in accuracy and other performance metrics.

### 4.4. Discussion

Multiple trials and comparisons are carried out to ensure that the proposed framework may boost the classification accuracy while keeping generalization. The model has been trained against a variety of data sets, and its efficacy is assessed using metrics like precision, recall, and F1-score. The results show that the hybrid model, which combines ViT with swarm optimization, outperforms existing approaches in terms of classification and generalization. These findings confirm the proposed model’s ability to extract the most beneficial features. From Figure 7, we can see that the proposed models record the highest results. VGG16 models report the lowest accuracy and ResNet18 models report the highest accuracy compared to pre-trained CNN models.

### 4.5. Comparison with the State of the Art

The comparison of the proposed ViT-PSO-SVM with the state of the art (SOTA) using SipakMed and Herlev datasets and different scenarios of classes: two, three, and five is shown in Table 10. There are related works that applied traditional methods of DL others use advanced models of DL such as ViT. In our work, ViT-PSO-SVM exhibits excellent performance, surpassing the other evaluated models in accuracy. These improvements can be attributed to the combination of multiple techniques. First the ViT for extracting local and global features then the utilization of the PSO to reduce feature complexity by selecting the most impacted features, and SVM for giving the last decision in the developed model. For SipakMed with five classes [18], ensemble model recorded 94.09 accuracy, and ResNet50 recorded 91.04 accuracy. ViT-PSO-SVM enhanced accuracy by 3%. For SipakMed with three classes [18], the CerviFormer-a model recorded 96.67 accuracies, ViT-PSO-SVM enhanced accuracy by 3%. For SipakMed with two classes [24], the CerviFormer-a model recorded 96.67of accuracy [51], ViT-PSO-SVM enhanced accuracy by 3%. Herlev with two classes [20,21,22], applied MLP classifier, CVM-Cervix, ViT-CNN and recorded 97.14, 97.14, and 97.65, respectively. ViT-PSO-SVM enhanced accuracy by 2%.

### 4.6. XAI Using GradCAM Heatmaps

The Sipakmed dataset comprises cervical cell images that are employed to evaluate and detect cervical precancerous and cancerous tumors. Those cervical cell images, tagged by various cellular features, provide a diverse set of data for combining XAI and GradCAM to better understand the AI model’s decision-making process. GradCAM can be utilized to create visual explanations for the model’s predictions. By superimposing the GradCAM heatmaps across the original cell images, GradCAM will highlight the regions of the input cell image that the model is concentrating on to determine its classification. To achieve this visualization, we utilize the PyTorch GradCAM package [54] to highlight the most significant regions of images that contribute to the class determination. Figure 8 shows GradCAM for each class. The following presents the details about each class.

Koilocytotic cells: Koilocytosis represents a cytopathic condition characterized by cells with a perinuclear halo, an uneven nuclear membrane, with a high nuclear-to-cytoplasmic ratio.Metaplastic cells: Metaplasia is the change of one cell type into another. Metaplastic cells, regarded as transitional cells between the original squamous epithelium and the newly created columnar epithelium, might be found in the cervix.Superficial cells: Superficial cells constitute mature, completely developed cells that reside in the cervical epithelium’s outermost layer.Dyskeratotic cells: Dyskeratosis is the abnormal keratinization of cells. Dyskeratotic cells have uneven cell shape, enlarged nuclear dimensions, with aberrant keratinization.Parabasal cells: Parabasal cells are immature basal-like cells that reside within the epithelium’s lower layers. The presence of them may indicate improper cell maturation.

## 5. Conclusions

A key factor in lessening the number of fatal cases of cervical cancer remains the early identification. This research introduces a novel approach named ViT-PSO-SVM, which combines the ViT Transformer, PSO feature selection, and SVM to enhance the classification performance. ViT Transformer is used for getting global and local pertinent data from images, making dependency detection simpler. The retrieved features are then processed by PSO for effective feature selection, allowing the model to focus on the most significant characteristics. SVM classifier is used in place of the SoftMax classifier with the objective of boosting classification accuracy and generalization capacity. Efficacy of ViT-PSO-SVM is compared with various models, including the ViT Transformer, CNN, and pre-trained CNN models, in extensive tests. The outcomes show exactly how significantly superior the suggested model is than the present versions. Metrics including accuracy, precision, recall, and F1-score are used to evaluate the model. It is important to note that the detection and classification are performed at the cell level, which may not fully capture the complexity of cancerous lesions at the tissue or organ level. ViT-PSO-SVM recorded the highest accuracy for two, three, and five classes on the SipakMed dataset as 99.112, 99.211, and 97.247, respectively, and 97.778 accuracy on the Herlev dataset. Finally, GradCAM heatmaps were generated as XAI criteria to visualize and understand the regions of an image that are important for the proposed model prediction. Future work should focus on expanding the dataset, exploring additional optimization techniques for the ViT-PSO-SVM model, conducting further comparative analyses, improving dataset collection and preprocessing techniques, and optimizing the model for real-world deployment.

## Figures and Tables

**Figure 1 bioengineering-11-00729-f001:**
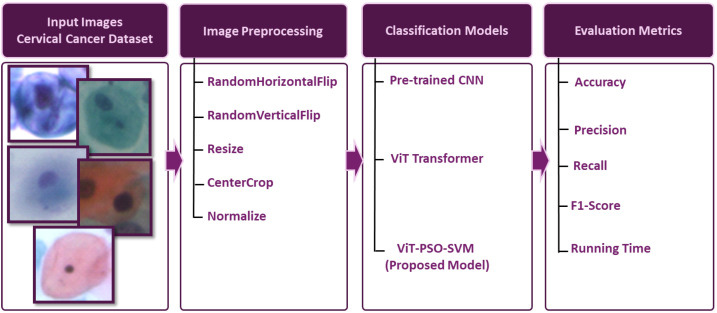
The proposed pipeline for classifying cervical cancer images.

**Figure 2 bioengineering-11-00729-f002:**
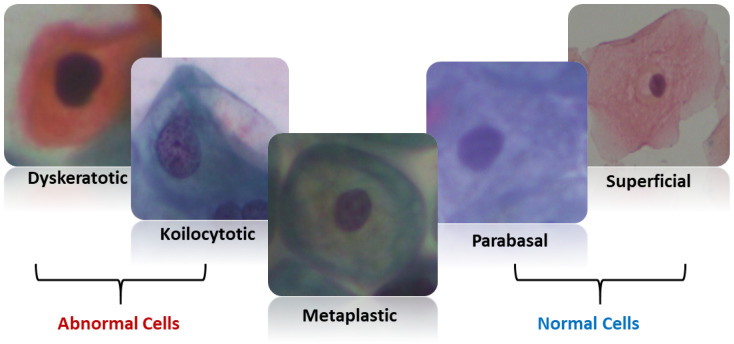
Example of images for SipakMed.

**Figure 3 bioengineering-11-00729-f003:**
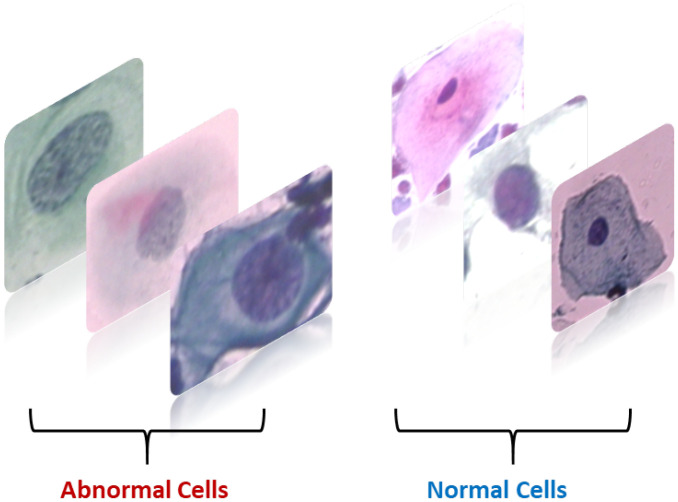
Example of images for Herlev.

**Figure 4 bioengineering-11-00729-f004:**
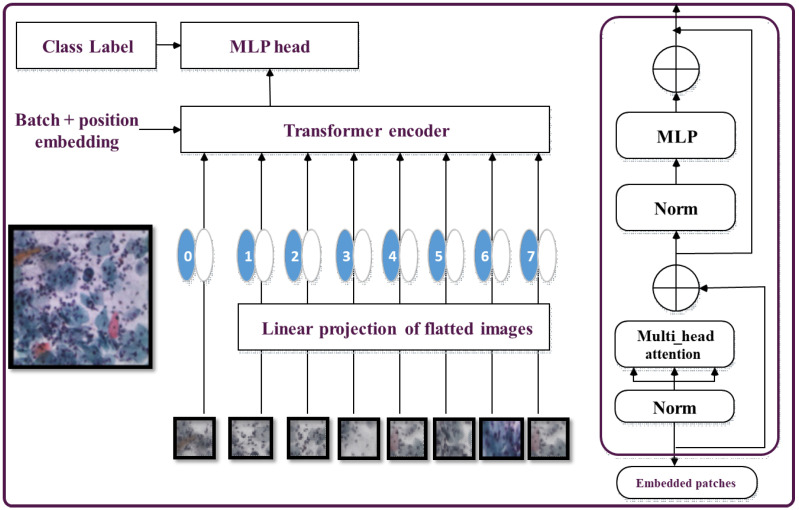
ViT Transformer model.

**Figure 5 bioengineering-11-00729-f005:**
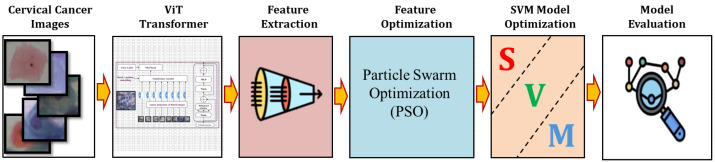
Blockdiagram of the proposed ViT-PSO-SVM model.

**Figure 6 bioengineering-11-00729-f006:**
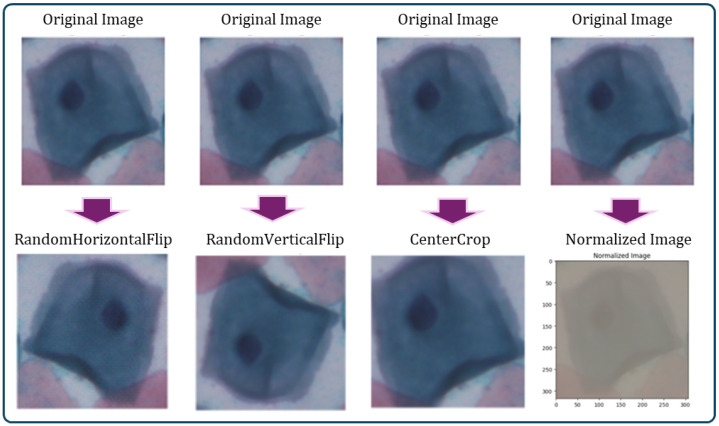
A visual representation of the effect of each preprocessing step on the images.

**Figure 7 bioengineering-11-00729-f007:**
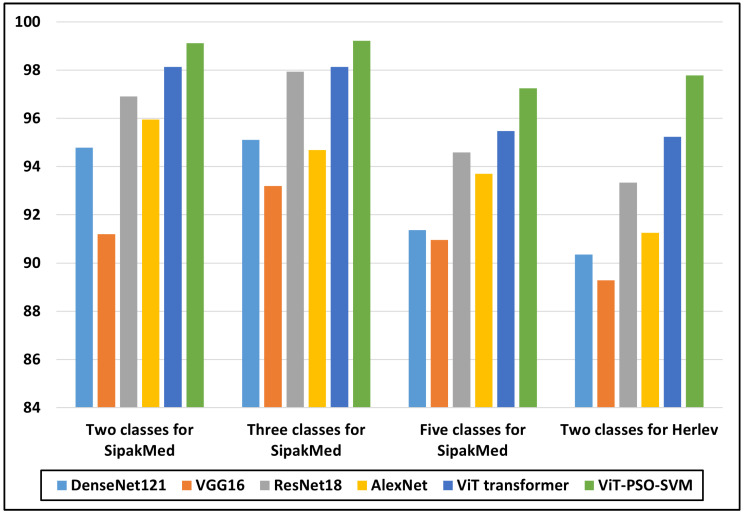
Graphical comparison between different models in terms of accuracy for SipakMed and Herlev datasets.

**Figure 8 bioengineering-11-00729-f008:**
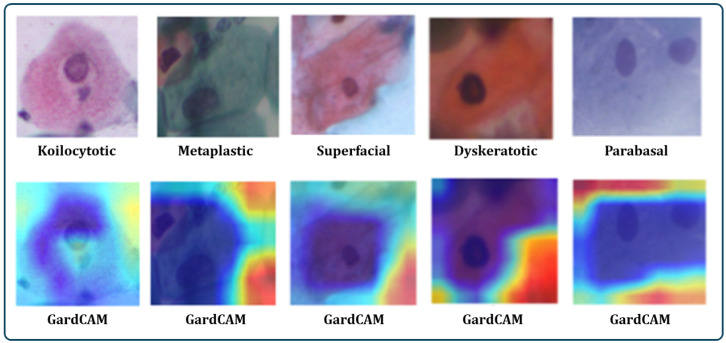
Visual representation of some Pap smear images using GradCAM heatmaps as an XAI technique for better insights of the proposed model prediction.

**Table 1 bioengineering-11-00729-t001:** The number of images in each class for SipakMed.

Five Classes	Three Classes	Two Classes	Training	Testing	Validation	Total
Dyskeratotic	Abnormal	Abnormal	569	204	40	813
Koilocytotic	Abnormal	Abnormal	577	207	41	825
Metaplastic	Benign	Abnormal	555	199	39	793
Parabasal	Normal	Normal	550	198	39	787
Superficial	Normal	Normal	581	209	41	831

**Table 2 bioengineering-11-00729-t002:** The number of images in each classes for Herlev.

Category	Training	Testing	Validation	Total
Normal	472	61	12	545
Abnormal	169	170	33	372

**Table 3 bioengineering-11-00729-t003:** Parameters of PSO.

Parameter	Value
Population size	20
Max num of generation	30
Early stopping	True
Local best weight	1
Global best weight	1
Use local random seed	True

**Table 4 bioengineering-11-00729-t004:** The size of extracted features after of optimized.

Datasets	Number of Classes	Features Size before PSO	Features Size after PSO	Trials
SipakMed	Five classes	800	219	7
	Three classes	800	250	5
	Two classes	800	242	4
Herlev	Two classes	800	233	2

**Table 5 bioengineering-11-00729-t005:** Image preprocessing/augmentation technique and its associated parameters.

Preprocessing Techniques	Parameter Value
RandomHorizontalFlip	0.5
RandomVerticalFlip	0.5
Resize	256
CenterCrop	224
Normalization	Mean and standardization

**Table 6 bioengineering-11-00729-t006:** Model performance of two classes using SipakMed.

Approaches	Model	Accuracy	Precision	Recall	F1-Score	Time
Pre-trained CNN	DenseNet121	94.787	94.923	94.787	94.736	1 h 43 m
	VGG16	91.200	91.200	91.200	91.200	2 h 15 m
	ResNet18	96.915	96.916	96.915	96.915	4 m 2 s
	AlexNet	95.957	95.957	95.957	95.938	3 m 34 s
ViT	ViT Transformer	98.126	98.127	98.126	98.124	4 m 8 s
**The proposed models**	**ViT-PSO-SVM**	**99.112**	**99.119**	**99.112**	**99.113**	5 m 13 s
	ViT-PSO-RF	98.915	98.918	98.915	98.914	5 m 11 s
	ViT-PSO-LR	98.521	98.531	98.521	98.518	5 m 12 s
	ViT-PSO-MLP	98.619	98.634	98.619	98.616	6 m 1 s

**Table 7 bioengineering-11-00729-t007:** Model performance of three classes.

Approaches	Model	Accuracy	Precision	Recall	F1-Score	Time
Pre-trained models	DenseNet121	95.112	95.115	95.112	95.113	2 h 34 m
	VGG16	93.199	93.187	93.199	93.157	4 h 20 m
	ResNet18	97.929	97.934	97.929	97.926	5 m 37 s
	AlexNet	94.688	94.896	94.688	94.744	5 m 40 s
ViT	ViT Transformer	98.126	98.133	98.126	98.126	6 m 41 s
**The proposed models**	**VT-PSO-SVM**	**99.211**	**99.211**	**99.211**	**99.211**	7 m 2 s
	VT-PSO-RF	99.102	99.102	99.102	99.102	7 m 5 s
	VT-PSO-LR	98.816	98.816	98.916	98.816	7 m 1 s
	VT-PSO-MLP	98.610	98.609	98.609	98.609	8 m 10 s

**Table 8 bioengineering-11-00729-t008:** Model performance of five classes using.

Approaches	Model	Accuracy	Precision	Recall	F1-Score	Time
Pre-trained CNN	DenseNet121	91.362	91.368	91.362	91.357	5 h 2 m
	VGG16	90.962	90.514	90.962	90.101	5 h 40 m
	ResNet18	94.592	94.733	94.592	94.586	5 m 33 s
	AlexNet	93.707	93.786	93.707	93.693	2 m 23 s
ViT	ViT Transformer	95.477	95.482	95.477	95.456	6 m 55 s
**The proposed models**	**ViT-PSO-SVM**	**97.247**	**97.253**	**97.247**	**97.239**	8 m 1 s
	ViT-PSO-RF	96.870	96.874	96.870	96.855	8 m 7 s
	ViT-PSO-LR	96.784	96.771	96.784	96.771	8 m 21 s
	ViT-PSO-MLP	96.870	96.861	96.870	95.853	8 m 8 s

**Table 9 bioengineering-11-00729-t009:** Model performance of two classes using Herlev.

Approaches	Model	Accuracy	Precision	Recall	F1-score	Time
Pre-trained models	DenseNet121	90.355	90.967	90.355	90.625	1 h 5 m
	VGG16	89.285	89.34	89.285	89.75	1 h 20 m
	ResNet18	93.333	93.889	93.333	93.002	1 m 30 s
	AlexNet	91.251	91.769	91.251	91.468	1 m 2 s
ViT	ViT Transformer	95.238	95.28	95.238	95.141	1 m 35 s
**The proposed models**	**VT-PSO-SVM**	**97.778**	**97.949**	**97.778**	**97.805**	2 m 20 s
	VT-PSO-RF	96.972	96.906	96.972	96.387	2 m 10 s
	VT-PSO-LR	96.372	96.347	96.372	96.357	2 m 3 s
	VT-PSO-MLP	96.342	96.203	96.342	96.193	2 m 30 s

**Table 10 bioengineering-11-00729-t010:** Comparison with the state of the art.

Study	Model	Number of Classes	Datasets	Accuracy
[18]	Ensemble	Two	Herlev	98.27
		Five	SipakMed	94.09
[52]	DenseNet-201	Two	Herlev	87.02
[20]	MLP classifier	Two	SipakMed	97.14
[21]	CVM-Cervix	Two	SipakMed	97.14
[22]	ViT-CNN	Two	SipakMed	97.65
[53]	ResNet50	Five	SipakMed	91.04
[23]	ResNet50	Two	SipakMed	97.5
[24]	CerviFormer-a	Three	SipakMed	96.67
[24]	CerviFormer-a	Two	Herlev	94.57
[25]	RNNS	Two	Herlev	87.75
**Proposed**	**ViT-PSO-SVM**	Two	SipakMed	**99.112**
		Three	SipakMed	**99.211**
		Five	SipakMed	**97.239**
		Two	Herlev	**97.778**

## Data Availability

All datasets used to support the findings of this study are publicly available and cited properly.

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
