# Peer review of "ViT-PSO-SVM: Cervical Cancer Predication Based on Integrating Vision Transformer with Particle Swarm Optimization and Support Vector Machine"

_bioengineering, 2024, doi:10.3390/bioengineering11070729_

Round 1
Reviewer 1 Report
Comments and Suggestions for Authors
This paper presents a novel diagnostic approach (ViT-PSO-SVM) for early prediction of cervical cancer (CCa) integrating Vision Transformer (ViT) with Particle Swarm Optimization (PSO) and Support Vector Machine (SVM). Utilizing benchmark datasets (SIPaKMeD and Herlev), the proposed model achieved high accuracy and F1-score, outperforming other models, demonstrating its potential as a robust, reliable, and non-invasive diagnostic tool.
Major comments
1. The study should incorporate an explanation of the model's performance using explainable AI (XAI) techniques, particularly by showing the attention layer of the ViT.
2. Please present the proposed ViT-PSO-SVM model with an illustration that is easy to understand, similar in style and clarity to Figure 4.
Minor comments
1. Figures 9, 10, 11, and 12 seem unnecessary.
Reviewer 2 Report
Comments and Suggestions for Authors
In this study, Vision Transformers (ViT), Particle Swarm Optimization (PSO) and Support Vector Machines (SVM) were integrated and a new method used in the diagnosis of cervical cancer (ViT-PSO-SVM) was developed. The proposed method has outperformed other traditional models by achieving high accuracy and F1 score on SIPaKMeD and Herlev datasets. This method has the potential to offer a non-invasive, reliable and sensitive diagnostic tool in the diagnosis of cervical cancer. But you need to make some changes in order for your work to be more understandable.
1- Make sure that the "SIPaKMeD" data set is written with the same spelling throughout the study.
2- How the image augmentation operations that you have provided in the image preprocessing section are applied are not explained in the experimental results section. Also, please explain how the number of images in the data has changed as a result of this application. Please share the parameters of the techniques you use for image preprocessing in your study.
3- The model proposed in the experimental results section (Swin-GA-RF) is different from the model proposed in the study (ViT-PSO-SVM). Check your work again.
4- Specify to what extent the PSO algorithm optimizes the features extracted from the normal ViT transformer model in your study. For example, did the PSO algorithm reduce from 200 features to 100 features? Make this clear in your work. How many times have you tried the PSO algorithm? Can you share the most optimal result and other trial results in your study?
5- Why is only SVM preferred from machine learning algorithms? Have you tried other machine learning algorithms such as random forest, neural network? Add the results to your study.
6- Add the block diagram of your study to the experimental results section. (ViT-PSO-SVM)
7- If you can give a graphical comparison in a single graph, it will be more understandable for readers. In the previous section, these results are given for each experiment in the confusion matrix.
8- Please include in your study the cost calculations of DenseNet121,VGG16, ResNet18, AlexNet, ViT transformer and ViT-PSO-SVM models.
Round 2
Reviewer 1 Report
Comments and Suggestions for Authors
I thank the authors for their efforts in revising the paper to incorporate my previous review comments.
Reviewer 2 Report
Comments and Suggestions for Authors
Thank you for the meticulous corrections you provided. Your detailed responses were thoroughly adequate.